# The Homologous Recombination Deficiency Scar in Advanced Cancer: Agnostic Targeting of Damaged DNA Repair

**DOI:** 10.3390/cancers14122950

**Published:** 2022-06-15

**Authors:** Vilma Pacheco-Barcia, Andrés Muñoz, Elena Castro, Ana Isabel Ballesteros, Gloria Marquina, Iván González-Díaz, Ramon Colomer, Nuria Romero-Laorden

**Affiliations:** 1Department of Medical Oncology, School of Medicine, Alcala University (UAH), Hospital Central de la Defensa “Gómez Ulla”, 28047 Madrid, Spain; vpacbar@mde.es; 2Department of Medical Oncology, Hospital Universitario Gregorio Marañón, 28007 Madrid, Spain; andresjesus.munoz@salud.madrid.org; 3Department of Medical Oncology, Instituto de Investigación Biomédica de Málaga (IBIMA), 29590 Málaga, Spain; elena.castro@ibima.eu; 4Department of Medical Oncology, Hospital Universitario La Princesa, 28006 Madrid, Spain; aisabel.ballesteros@salud.madrid.org (A.I.B.); rcolomer@seom.org (R.C.); 5Department of Medical Oncology, Department of Medicine, School of Medicine, Complutense University (UCM), Hospital Universitario Clínico San Carlos, IdISSC, 28040 Madrid, Spain; gloria.marquina@salud.madrid.org; 6Department of Obstetrics and Gynecology, Hospital Universitario Severo Ochoa, 28911 Madrid, Spain; ivan.gonzalez@salud.madrid.org

**Keywords:** homologous recombination deficiency, HRD, *BRCA*, PARP inhibitors, agnostic cancer

## Abstract

**Simple Summary:**

Tumor-suppressor genes are involved in DNA break repair through the homologous recombination system and are widely known for their role in hereditary cancer. Beyond breast and ovarian cancer, prostate and pancreatic cancer also have targetable homologous recombination deficiency (HRD) beyond the well-known *BRCA1* and *BRCA2* with relevance that exceeds diagnostic purposes. In this review, we aim to summarize the roles of HRD across tumor types and the treatment landscape to guide the targeting of damaged DNA repair based on the cancer’s genetic features.

**Abstract:**

*BRCA1* and *BRCA2* are the most recognized tumor-suppressor genes involved in double-strand DNA break repair through the homologous recombination (HR) system. Widely known for its role in hereditary cancer, HR deficiency (HRD) has turned out to be critical beyond breast and ovarian cancer: for prostate and pancreatic cancer also. The relevance for the identification of these patients exceeds diagnostic purposes, since results published from clinical trials with poly-ADP ribose polymerase (PARP) inhibitors (PARPi) have shown how this type of targeted therapy can modify the long-term evolution of patients with HRD. Somatic aberrations in other HRD pathway genes, but also indirect genomic instability as a sign of this DNA repair impairment (known as HRD scar), have been reported to be relevant events that lead to more frequently than expected HR loss of function in several tumor types, and should therefore be included in the current diagnostic and therapeutic algorithm. However, the optimal strategy to identify HRD and potential PARPi responders in cancer remains undefined. In this review, we summarize the role and prevalence of HRD across tumor types and the current treatment landscape to guide the agnostic targeting of damaged DNA repair. We also discuss the challenge of testing patients and provide a special insight for new strategies to select patients who benefit from PARPi due to HRD scarring.

## 1. Introduction

Targeting homologous recombination deficiency (HRD) has been revealed in the last few years as one of the most promising strategies for various types of tumor. Alteration in the homologous recombination system (HR) genes is prevalent across tumor types and can be mostly found in breast, ovarian, pancreatic and prostate cancer [1,2]. HRD secondary to pathogenic germline and somatic variants in HR-associated genes has been reported as a predictive biomarker to inform about tumor sensitivity to platinum-based regimens and poly-ADP ribose polymerase (PARP) inhibitors (PARPi) [3,4]. Moreover, there are secondary changes to genetic mutations in DNA that can be detected at a structural level as genomic instability and could be associated with HRD response biomarkers that have already been validated. Identifying patients that may respond to direct or indirect therapies against HRD is a need in current clinical practice for cancer management and could optimize the clinical benefits of these therapies.

## 2. Homologous Recombination: A Key Pathway in DNA Repair

*BRCA1* and *BRCA2* are the best-known proteins involved in double-strand DNA break repair by HR. They are two of the main characters in the DNA defect situation, as shown by hundreds of publications reported in recent years [5]. However, genetic and epigenetic inactivation of other HR components can lead to HRD in sporadic cancers, classically termed *BRCA*ness [6]. HR is responsible for repairing DNA before the cell comes into mitosis. It is produced during and immediately after DNA replication in S and G2 phases of the cell cycle, when sister chromatids are available [7]. Double-stranded breaks induced by ionizing radiation or toxic agents as chemotherapy are first sensed by the MRE11-RAD50-NBN (MRN) complex, which loads helicase and exonucleases onto the breaks to start 5′–3′ double-stranded DNA resection. ATR then localizes to the ssDNA ends and switches on the ATR-dependent checkpoint, arresting the cell cycle for HR to proceed. Next, *BRCA1* is phosphorylated in response to DNA damage by DNA-damage response kinases, such as *ATM*, *ATR* and *CHK1*, which enables the cell to repair DNA before entering mitosis and survive [8,9,10,11]. *ATM*, *ATR*, *BARD1*, *RB*, *p53*, *p21* and their downstream effectors are involved in induced G1/S arrest [12]. Therefore, *BRCA1* loss can result in defective S-G2/M and spindle checkpoints that together with abnormal centrosome duplication and defective DNA damage repair can lead to genetic instability [12]. Furthermore, *BRCA2* can help to protect telomere integrity loading *RAD51* during S/G2 [7,13]. Proteins involved in the HR system have functions in DNA repair, but also participate in cell cycle regulation, transcriptional activation and chromatin remodeling (Figure 1).

Cancer genomics often harbor chromosomal aberrations arising from a defective HR pathway. In *BRCA* mutant cells, chromosomal spreads reveal increased gross chromosomal rearrangements [14]. This leads to the development of assays to evaluate the “genomic scar” left behind by the loss of HR function, irrespective of which component of the pathway was lost.

## 3. Prevalence and Prognostic Value of HRD in Cancer

Heritable damage in the DNA repair system can be observed in up to 10% of cancer patients. *BRCA1* and *BRCA2* are the most common of these genetic abnormalities, and breast-ovarian syndrome is the classical phenotype of germline *BRCA* alteration [1]. However, in the last decade, evidence has shown that somatic events are more frequent than previously expected and that these aberrations affect tumors beyond breast and ovarian cancer. Somatic mutations in *BRCA* genes are more frequent in ovarian cancer (15%), followed closely by prostate cancer, squamous skin cell carcinoma, breast cancer (around 10%) and pancreatic cancer. (Figure 2). It is noted that frequency varies significantly depending on the population studied, geographic area, type of sample studied or stage.

Other relevant HR pathway members include genes such as *ATM*, *PALB2*, *CHECK2* and *RAD51*. However, the associations between these genes and an HRD phenotype may be less consistent than those for *BRCA1* and *BRCA2* and may vary by the tumor’s tissue of origin [15,16,17]. Norquist et al. [17] observed that 6.8% of the ovarian cancer patients included in the GOG 218 trial harbored a non-*BRCA* somatic HR gene mutation, and the most frequently observed alteration was in *ATM*. The most frequently altered DNA-repair genes in both germline and somatic cells of mCRPC patients are *BRCA2*, *ATM* and *CHEK2*: germline mutations are found in 5%, 2% and 2%, respectively [18,19]. *BRCA*1/2 homozygous deletions are infrequent except in prostate cancer, where *BRCA*2 deletions have been reported at 2.6% frequency and accounted for 25% of *BRCA*1/2-altered cases [20].

HRD prevalence based on the measurement of telomeric allelic imbalance (TAI), loss of heterozygosity (LOH) and large-scale state transitions (LST) is the more extended diagnostic method to measure HR status and varies also broadly among different types of tumors, although there is a correlation with *BRCA* prevalence. TAI, LOH and LST are highly correlated with each other and reflect increasing genomic instability. TAI refers to allelic imbalance extending to the subtelomeric region >11 megabases (Mb) in size. LOH refers to permanent loss of one parent’s contributed allele copy at a specific locus, leading to homozygosity at that genomic site. LSTs refer to allelic imbalance > 10 Mb in size between adjacent genomic regions due to translocations or copy gains/losses.

The combination of these three parameters of genomic instability provides HRD scores by measuring LOH, TAI and LST with somatic next-generation DNA sequencing and may estimate the underlying genomic scarring in the context of HR deficiency. HRD scores have been more extensively studied for ovarian cancer, but when the algorithm is applied across different tumor types, it should be considered that some modifications are applied to avoid bias, and there is still no consensus [21]. Thus, this is not a validated method to be used in clinical practice for agnostic tumors [15]. Ovarian cancer, followed by lung adenocarcinoma and breast cancer, are the tumors where these alterations are more frequent, having genomic scores higher than 30 (Figure 3). In an in silico analysis of 5371 tumors of 15 cancer types available in the TCGA, cancers where platinum constitutes standard first-line therapy showed increased genomic scar scores.

Lotan et al. [15] evaluated HRD scores in prostate cancer and their associations with HR gene mutations, and observed that HRD scores vary significantly between patients harboring *BRCA2*, *ATM* and *CHEK2*. Germline *BRCA2*-altered prostate cancer patients had the highest HRD scores, germline *ATM*-altered patients had intermediate scores and germline *CHEK2*-altered patients had the lowest scores [15].

The most common genomic scar assays reported to date are two commercially available tests that combine tumor *BRCA* mutation testing with a Genomic Instability Score (GIS) based on quantification of TAI, LOH and LST. These tests are myChoice^®^ HRD test (Myriad Genetics, Salt Lake City, UT, USA) and Foundation Focus CDx*BRCA* HRD^®^ (Foundation Medicine, MA, USA).

Ovarian cancer has the strongest association with HRD, and up to 50% of high-grade serous ovarian carcinoma have a genetic aberration in the HR pathway [25]. In general, somatic aberrations are twice as frequent as germline alterations [20]. In non-endometrioid TP53-mutant endometrial cancer, which is molecularly similar to high-grade serous ovarian carcinoma, a high incidence of HRD genomic scars of up to 48% has been reported [26]. The prognostic significance of HRD in ovarian cancer is controversial. It has been reported to have more favorable overall survival (OS) compared to non-carriers [27], for both *BRCA1* (HR 0.78; 95% CI, 0.68–0.89; *p* < 0.001) and *BRCA2* mutation carriers (HR 0.61; 95% CI, 0.50–0.76; *p* < 0.001). However, Candido-dos-Reis et al. [28] analyzed the effect of germline *BRCA* mutations in 4314 ovarian cancer patients with a 10-year follow-up and showed that the better short-term survival observed decreased over time, and patients who harbored a *BRCA1* mutation even showed worse OS. Mutations in non-*BRCA* HR genes, including *ATM*, *CHEK2*, *PALB2* and *RAD51c*, have been reported to be predictors of survival in ovarian cancer patients [16].

The prognostic relevance of *BRCA* in breast cancer is also questionable: some studies demonstrated that patients with a *BRCA1/2* mutation had worse OS [29,30,31,32], and other studies showed no significant differences when compared with non-carriers [33,34,35,36]. Patients diagnosed with HRD breast cancer have shown an association with a more aggressive phenotype: *BRCA1* is more frequently associated with triple-negative breast cancer, and *BRCA2*-related breast cancer correlated with a higher histological grade compared to patients who do not have germline mutations [37,38,39]

The prognostic significance of HRD in patients with pancreatic adenocarcinoma is currently unknown [40]. Golan et al. [40] analyzed 71 patients with *BRCA1/2*-asssociated pancreatic cancer and observed an improvement in survival in patients with advanced disease (stage 3 and 4) who had received platinum-based therapy in comparison to those patients who were not treated with these agents. In their study, a more favorable outcome with platinum treatment was suggested, but a statistically significant improvement was not observed [40].

In prostate cancer, germline *BRCA2* mutations have been associated with a more aggressive phenotype and poorer outcomes [41]. Castro et al. [42] studied germline DNA repair defects in an unselected cohort of patients with metastatic castration-resistant prostate cancer and observed that *gBRCA2* mutation was an independent prognostic factor for cause-specific survival in this setting. The prognostic role of somatic *BRCA2* alterations remains unclear.

## 4. HRD as an Actionable Target

The treatment landscape has evolved in the last decade, and HRD has been proposed as a predictive biomarker to determine increased sensitivity to platinum chemotherapy and PARPi [43,44].

### 4.1. Platinum

Platinum chemotherapy binds directly to the DNA in order to cause the cytotoxic effect of crosslinking DNA strands and induce double-stranded breaks, which are not repaired in cells that harbor defects in involved DNA repair pathways. In the last decade, *BRCA* mutations, as lead actors of HRD, have been suggested as predictive biomarkers for response to platinum across different tumor types [45,46,47,48,49,50,51,52] (Table 1). However, despite of high rates of platinum sensitivity in this population, it seems much more is needed beyond *BRCA* alterations to select candidates for treatment.

In ovarian cancer, the standard first-line chemotherapy regimen includes platinum and taxanes, independently of *BRCA* status, and response rates are greater than 80%. Platinum-free interval has been identified as a key biomarker to response to subsequent lines. Higher intervals are associated with predominance of HRD in tumor progression, and this will determine the indication for platinum re-treatment.

In breast cancer, an increasing amount of evidence suggests that TNBC patients with *BRCA* mutations could be more sensitive to platinum-based chemotherapy [46,53]. A recently published meta-analysis by Chai et al. [54] included six trials with HRD in TNBC data and showed that patients with HRD-positive TNBC had higher complete response rates compared to HRD-negative ones after receiving platinum-based neoadjuvant chemotherapy. However, the GeparSixto trial showed that the response to platinum-agents was not dependent on *BRCA* status [47], and TNBC non-mutated *BRCA* patients showed increased response rates with carboplatin, meaning that HRD did not predict carboplatin benefit [55].

HRD associated pancreatic cancer was under-identified until recently [56]. A family history of breast, ovarian or pancreatic cancer has been associated with increased sensitivity to platinum drugs as DNA-damaging agents [57], suggesting the presence of DNA repair defects in those patients. However, studies considering only clinical inheritance have failed to demonstrate this clinical biomarker as effective. Thus, Okano et al. [49] evaluated platinum benefit in patients with a family history of ovarian, prostate, pancreatic or breast cancer without analyzing *BRCA* genes and did not observe a benefit in OS. Patients with HRD pancreatic cancer have shown a clinical benefit and a longer OS due to platinum-based treatments [50,58,59,60], and data also suggest a notable and significant increase in the response rates: 50–65% [61]. Similar activity of oxaliplatin and cisplatin in patients with germline *BRCA* and *PALB2* mutations has been suggested by retrospective data [50,62], and a survival benefit of platinum-based first line chemotherapy in this subgroup of patients have been observed [50,63].

Metastatic prostate cancer patients harboring DNA repair gene alterations treated with platinum-based chemotherapy showed encouraging antitumor activity [51,52], although the role of HRD in this setting is still controversial [51,52]. Recently, Pokataev et al. [64] published a meta-analysis reporting higher overall survival in patients with HRD, advanced prostate cancer treated with platinum-based chemotherapy. Prospective validation in ongoing randomized clinical trials will be needed to determine the role of platinum treatment in advanced prostate cancer.

### 4.2. PARP Inhibitors

In 2005, two groundbreaking studies observed that tumor cells lacking *BRCA1* or *BRCA2* were particularly sensitive to PARPi through various mechanisms [66,67]. The main target of PARPi is *PARP1*, which is involved in the repair of single-strand DNA breaks, so in order to produce cytotoxicity, a defective HR is required [66,67].

*PARP1* is a damage sensor that is able to synthesize PAR chains on target proteins near DNA break, and with these PAR chains recruit additional DNA repair effectors [5]. PARPi causes a catalytic inhibition of *PARP1* and traps *PARP1* by either inhibiting autoPARylation or by causing allosteric changes in its structure [68,69]. Patients’ tumors which lack *BRCA1* or *BRCA2* are not able to repair DNA lesions and try to use error-prone DNA repair pathways that have a cytotoxic effect that kills the cells [5].

In 2015, a basket trial of olaparib in patients with *gBRCA1/2* mutations identified responding patients beyond the ovarian or breast cancer population, suggesting that other HR-defective tumors could be suitable for PARPi treatment [70]. The current developments of PARPi in solid tumors are displayed in Table 2.

Three PARPi, olaparib, rucaparib and niraparib, have been approved by the United States Food and Drug Administration (FDA) and the European Medicines Agency (EMA) as maintenance therapy in platinum-sensitive recurrent epithelial ovarian cancer [71,72,73,74]. Rucaparib has been also approved as monotherapy in patients with somatic or germline *BRCA1/2* mutations [74]. In 2018, the FDA granted approval to olaparib monotherapy for the first-line maintenance treatment of patients with *BRCA*-mutated advanced epithelial ovarian, fallopian tube or primary peritoneal cancer who have completely or partially responded to first-line platinum-based chemotherapy based on SOLO-1 trial results [75]. Shortly after, niraparib achieved indication for same setting independently of *BRCA* status based on a PRIMA trial [76]. Analysis subgroups revealed a benefit for all populations, although less significant for HRD-proficient patients. The PAOLA-1 phase III first-line ovarian cancer maintenance study showed a benefit on progression free survival of the combination of olaparib and bevacizumab compared to bevacizumab in the overall population; however, this benefit was not seen in the subgroup analysis of the HRD-proficient population [77]. On May 2020, the FDA expanded the approval of olaparib and bevacizumab for first-line maintenance treatment of HRD, advanced ovarian cancer [78].

Metastatic breast cancer patients with a germline *BRCA1* or *BRCA2* mutation treated with PARPi have better outcomes in terms of PFS compared to standard chemotherapy [79,80]. Recently, the OlympiA trial evaluated PARPi efficacy in early breast cancer after standard adjuvant treatment with chemotherapy and local therapy, achieving in patients with *gBRCA1/2* mutations longer survival, free of invasive or distant disease, than the placebo [81].

In metastastic *gBRCA*-mutated pancreatic adenocarcinoma, olaparib has been approved for the maintenance after at least 16 weeks of first-line platinum-based chemotherapy if the disease has not progressed [82]. The objective response rate was 23% in the olaparib arm; 10% for placebo and 10% of patients from the placebo arm maintained a median duration of response of 24.9 months. On the other hand, cisplatin plus gemcitabine was evaluated in a phase II trial of 50 patients with *gBRCA* or *PALB2*-mutated locally advanced or metastastic pancreatic cancer randomly assigned alone or with veliparib [59], and concurrent veliparib did not improve the response rate in this subset of patients.

Several PARPi are currently under development for the treatment of advanced prostate cancer [83,84,85,86,87,88]. Alterations in *BRCA2*, particularly homozygous deletions, seem to be the best predictor of response to PARP inhibition [89]. In patients with *BRCA1/2* alterations, 40–46% radiographic response rates have been reported with the different agents. PSA declines of >50% have been noted in half of the *BRCA1/2* patients included in the different trials, despite being heavily pretreated. No differences in efficacy have been reported based on the germline or somatic origin of the alterations [87]. The predictive roles of other HR defects beyond *BRCA1/2* remain unclear. Little or no benefit from PARP inhibition has been observed in patients with *ATM* or *CDK12* alterations, whilst the predictive roles of less frequent alterations have not been stablished due to the limited number of patients included in trials. Olaparib has been the only PARPi to be investigated in monotherapy in a phase 3 trial for advanced prostate cancer patients. In the PROfound study, men with metastatic castration-resistant prostate cancer (mCRPC) with alterations in one of the 15 HR genes screened whose disease had already progressed to an AR-targeting inhibitor (ARTi) were randomized to receive treatment with olaparib 300 mg bid or a second ARTi. A benefit in overall survival was observed for patients in cohort A, which included patients with *BRCA1*, *BRCA2* and *ATM* alterations, whereas no benefit was observed for patients with other alterations included in cohort B. These results led to the EMA approval of olaparib for the treatment of mCRPC patients with *BRCA1/2* alterations after disease progression to treatment with an ARTi.

In non-small lung cancer and colorectal cancer, two of the most prevalent tumors, preliminary in vivo studies in cell lines with HRD features have supported the potential use of PARPi [90,91].

Uterine leiomyosarcoma has recently been identified as a sarcoma subtype with characteristic defects in the HR repair pathway and frequent *BRCA2* loss [92]. Preclinical data demonstrate marked activity for PARPi in combination with the alkylating agent temozolomide. Ongoing research in order to identify other sarcomas with DNA repair defects is promising, and may offer a new opportunity for the targeted treatment of this rare, aggressive cancer [92].

## 5. The Challenge of Testing: Searching for HRD Scar

Next generation sequencing (NGS) techniques, as the current gold standard of genetic diagnosis, have helped to shorten the time to obtain a genetic test result. However, at the same time, an NGS multi-panel may provide data about other HRD genes beyond *BRCA* where the current evidence as biomarkers to select therapy is limited, hindering decision making in clinical practice. To handle this complexity, an adequate bioinformatic analysis will be key, along with a multidisciplinary approach.

Traditionally, *BRCA* testing has been conducted in germline DNA triggered by a familial aggregation of cancer. Recent studies have demonstrated that a significant proportion of mutation carriers are undiagnosed due to the lack of a significant family history of cancer [18,42,98], leading to changes in the recommendations for genetic testing. As an example, testing is now recommended for all metastatic prostate cancer patients, regardless of their personal or family history of cancer [99]. Moreover, the advent of therapies that target *BRCA* alterations and other HRD defects requires the investigation of germline mutations and alterations acquired by the tumor, as described previously.

For that reason, somatic mutation analysis is moving germline testing in various scenarios, such as advanced ovarian and prostate cancer. HRD testing of the tumor directly has the advantage of providing higher rates of positivity compared with germline tests [19]. However, somatic testing is associated with higher rates of failure for sequencing [84]. For that reason, new protocols for improving the conservation and storage of paraffined samples should be implemented in hospitals. In fact, a systematic and consensus protocol for high-quality minimum biomarker testing is being requested by the scientific community [100]. These molecular testing recommendations should be offered to all cancer patients for diagnosis and prevention, detailing the type, the technique and the methods of implementation and ensuring adequate training for clinicians to guide the treatment decisions.

ESMO and NCCN guidelines [98,101,102] recommend pre-test counseling to determinate the most appropriate test for each patient and specific post-test counseling when results are available. Table 3 summarizes the current recommendations for testing based on international clinical guidelines.

A different strategy to identify HRD patients could be to measure the “genomic scarring” associated with loss of function in DNA repair pathways, as genomic instability. In 2012, three SNP-based assays were developed to quantify the extent of chromosomal abnormalities related to HRD: (a) TAI due to inappropriate chromosomal end fusions because of aberrant end joining; (b) LOH—related to inaccurate repair of sister chromatids during the S/G2 cell cycle phase; and (c) LST—chromosomal breaks of more than 10 Mb. These “functional assays” have been proposed as more reliable methods for identifying patients responding to PARPi compared to simply identifying gene mutations, although their role in guiding therapy is pending on validation, and their values and thresholds are heterogeneous across cancer types [103].

In 2020, the phase III PAOLA-1 trial in ovarian cancer was the first one to obtain FDA/EMEA approval for HRD-positive patients to use the PARPi combination with bevacizumab using the Myriad myChoice^®^ test defined by GIS ≥ 42 [77]. However, the evaluations of the utility of these tests to predict the benefit from PARPi in the *BRCAwt* populations were preplanned secondary analyses in clinical trials, and there are not definitive results for trials specifically designed for this subpopulation [103,104]. In breast cancer patients, GIS ≥ 42 has been associated with *BRCA* mutations, and furthermore, TAI has been associated with an improved response to platinum chemotherapy [105,106]. In pancreatic cancer patients, GIS ≥ 42 has shown a sensitivity of 91% and a specificity of 83% for the identification of HRD. Moreover, a higher GIS has been associated with an improved oncological outcome with platinum chemotherapy [107]. Currently, there are different thresholds proposed as best classifiers for HRD score evaluation, but they are still pending validation [108].

In prostate cancer, non-commercial signatures based on genomic instability scores have been explored. An adequate correlation between *BRCA*-deficient samples and HRD-associated mutational signatures using WGS data was reported [109]. However, there are not clinical trials demonstrating the optimal threshold to assure the role of the HRD score testing as a predictor of treatment with PARPi or platinum therapy.

Another type of functional assay that may have the potential to provide a dynamic readout of HRD scarring is based on the estimation of the amount of nuclear *RAD51*, a downstream HR protein (a DNA recombinase). *RAD51* enables high-fidelity double-strand DNA repair by facilitating DNA strand invasion into the sister chromatid, a process supported by the *BRCA1/PALB2/BRCA2* complex. Reduced, DNA-damaged, induced nuclear RAD51 foci have been associated with *BRCA1* or *BRCA2* gene defects and PARPi responses [110,111]. This approach is currently under investigation, and functional assays have not yet been validated for pancreatic cancer [112].

Additionally, the HRD profile may change during cancer progression, as reversion mutations of HR genes have been reported to occur in 26% of patients, and this fact may be related to response to previous treatment to generate resistance to platinum or PARPi [113]. In this setting, monitoring the dynamic evaluation of HRD in cancer should be relevant. Moreover, one of the main difficulties in tumors such as pancreatic cancer is obtaining an adequate amount of tissue for genetic testing. Thus, liquid biopsy approaches to identify HRD based on circulating tumoral DNA analysis to assess chromosomal instability or mutational signatures is a promising method under study, but is not ready yet to use in clinical practice (pending validation) [114].

## 6. Conclusions

The detection of DNA repair defects related to the HR pathway provide a unique opportunity for the development of treatments in different type of tumors that take advantage of a same tumor feature. Tools and validation trials to identify the optimal HRD test across tumor types are urgently needed.

## Figures and Tables

**Figure 1 cancers-14-02950-f001:**
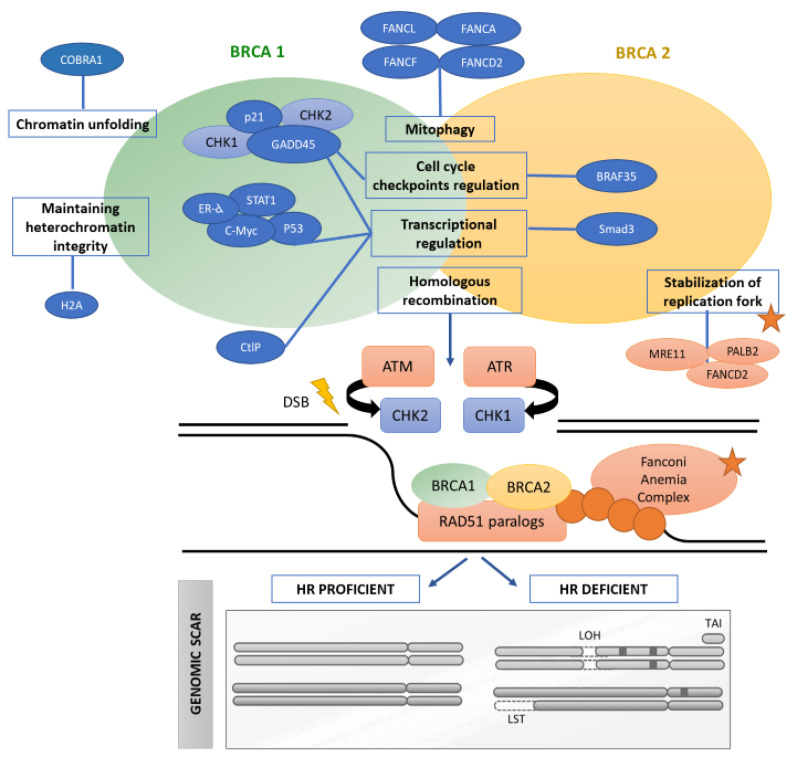
Functional features of proteins involved in HRD. Adapted with permission from Gorodetska et al. [7], copyright 2019 the authors, Ivyspring International Publisher under the terms of the Creative Commons Attribution license 4.0 (https://creativecommons.org/licenses/by-nc-nd/4.0/, accessed on 12 May 2022) and Hoppe et al. [6], copyright 2018 by the authors published by Oxford University Press. The figures have been modified for the purposes of this review. Proteins involved in the HR system have functions in DNA repair, but also participate in cell cycle regulation, transcriptional activation and chromatin remodeling. Genomic scarring is defined by the presence of chromosomal abnormalities related to HRD: (a) telomeric allelic imbalance (TAI) due to inappropriate chromosomal end fusions because of aberrant end joining, (b) loss of heterozygosity (LOH) related to inaccurate repair of sister chromatids during the S/G2 cell cycle phase and (c) large-scale transitions (LSTs) that are chromosomal breaks of more than 10 Mb.

**Figure 2 cancers-14-02950-f002:**
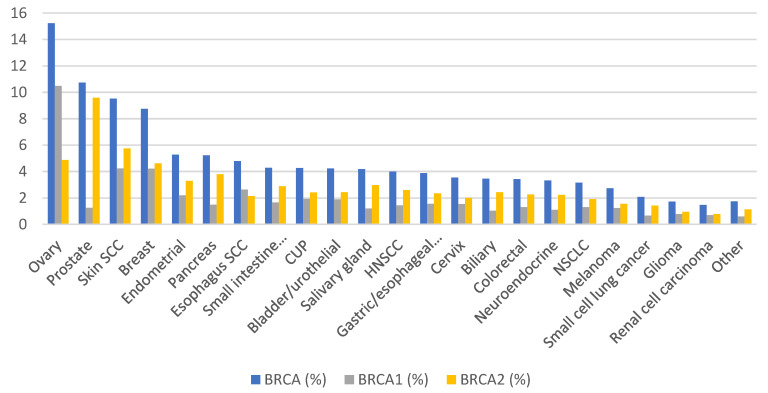
Prevalence of somatic *BRCA1/2* mutations across different tumor types. Adapted from Sokol et al. [20], copyright 2020 the authors, American Society of Clinical Oncology under the Creative Commons Attribution Non-commercial No Derivatives 4.0 License (https://creativecommons.org/licenses/by-nc-nd/4.0/, accessed on 12 May 2022). The figure has been modified for the purposes of this review.

**Figure 3 cancers-14-02950-f003:**
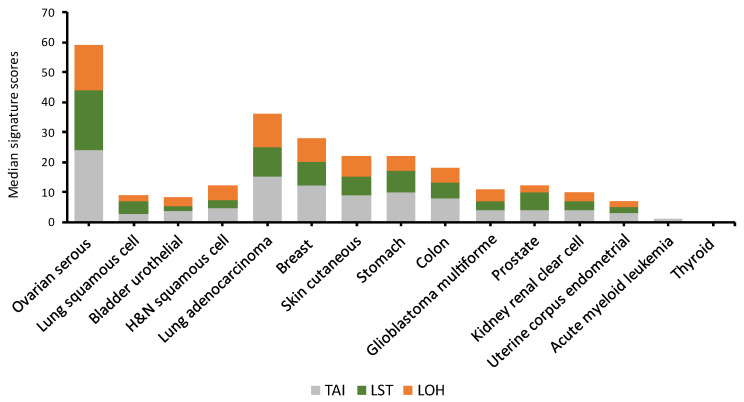
HRD prevalence across different tumor types. Adapted from Marquard et al. [21], copyright 2015 Marquard et al, under the Creative Commons Attribution Non-commercial No Derivatives 4.0 License (https://creativecommons.org/licenses/by-nc-nd/4.0/, accessed on 12 May 2022). The figure has been made for the purposes of this review. HRD analysis of TCGA samples across 15 different cancer types was performed based on the number of Telomeric Allelic Imbalances (TAI) based on a genomic scar accumulation, the large scale transition (LST) based on a type of genomic scar associated with loss of *BRCA1* or *BRCA2* and the HRD-LOH based on a scar enriched in high-grade serous ovarian cancer patients with a loss of *BRCA1* or *BRCA2* [21,22,23]. However, the method originally used for ovarian cancer samples was adapted to avoid bias when the algorithm is applied across different tumors: (1) In the original publication describing TAI [24], all allelic imbalance events that extended to the telomere were counted, if they did not span the centromere. This results in an overrepresentation of tumors with an uneven copy number among high TAI cases, which has been corrected in the method used for the present study. (2) The original publication describing HRD-LOH [23] excluded chromosome 17 because LOH on chromosome 17 in the ovarian cancer samples is ubiquitous and for this reason did not provide independent information. However, for this figure, chromosome 17 was not excluded, as chromosome 17 is not ubiquitously lost in all cancer types, and therefore may provide independent information in some tumor samples.

**Table 1 cancers-14-02950-t001:** Clinical trials evaluating platinum therapy for HRD tumor types.

Type of Tumour	Author	Type of Study	N	Primary Endpoint	Platinum	Benefit	Target	Sub-Population
Breast Cancer								
Localized	Tung 2020 [45]	Randomized Phase II	118	pCR	Cisplatin	Platinum vs. Control: 18% vs. 26%. Risk ratio 0.70 (90% CI, 0.39–1.2).	HER2- I-III	69% *gBRCA1* 30% *gBRCA2* 2% Both
	Hahnen 2017 [47]	Randomized Phase II	50	pCR	Carboplatin	Platinum vs. Control: 65.4% vs. 66.7%. Odds ratio 0.94 (0.29–3.095), (*p* = 0.92).	TNBC II-III	17% *gBRCA1/2*
Advanced	Isakoff 2015 [65]	Phase II	86	ORR	Cisplatin Carboplatin	*BRCA1/2* mut vs. wild type: 54.5% vs. 19.7%, (*p* = 0.022).	TNBC metastatic or locally recurrent unresectable	13% *gBRCA1/2* 77% wild type 10% not known
Pancreatic Cancer								
Localized	Golan 2020 [48]	Retrospective analysis	61	pCR	Oxaliplatin	Mutated vs. non-mutated: 44.4% vs. 10%, (*p* = 0.009).	Borderline resectable	23% *gBRCA2* 77% *gBRCA* wild type
Metastatic	Okano 2020 [49]	Phase II	43	OS	Oxaliplatin	1-year survival 27.9% (90% CI 17–41.3). Primary endpoint not met (30%).	Metastatic PDAC	Family history (ovarian, prostate, pancreatic, breast) - *BRCA* not known
	Wattenberg 2020 [50]	Retrospective analysis	26	PFS	Oxaliplatin Cisplatin	Mutated vs. non-mutated: 10.1 vs. 6.9 months, (*p* = 0.0068)	Locally advanced or metastatic	33% Mutated: - 19.2% gBRCA1 - 65.4% gBRCA2 - 15.4% gPALB2 - 67% Non-mutated
Prostate Cancer								
Castration-resistant prostate cancer	Schmid 2020 [51]	Retrospective analysis	508	Platinum Antitumor activity (decrease PSA 50% and/or radiological response)	Carboplatin Cisplatin Oxaliplatin	Mutated (cohort 1) vs. non-mutated (cohort 2) decrease PSA: 47.1% vs. 36.1%, (*p* = 0.20).	Advanced	- 15.7% Mutated (cohort 1): - 55% *BRCA2* - 15% *ATM* - 3.8% *BRCA1* - 19.3% Non-mutated (cohort 2) - 65% Unknown (cohort 3)
	Mota 2020 [52]	Retrospective analysis	109	Platinum efficacy in DDR-mutant	Carboplatin Cisplatin	67% *BRCA2* achieved a PSA50 response (adjusted Odds Ratio 9.5; 95% CI 1.5–82.9) compared to DDRwt (13%), (*p* = 0.022).	Metastatic	- PARPi naïve and prior taxane: - 9% *BRCA2* - 3% *ATM* - 6% *CDK12* - 6% *FANCA* - 1% *PALB2* - 75% DDRwt

pCR: Pathologic complete response; ORR: objective response rate; OS: overall survival; PFS: progression free survival; CI: confidence interval. g: germline mutation. PSA: prostate-specific antigen. DDR: DNA damage repair, including somatic and germline mutations in *BRCA1/2*, *ATM*, *CDK12*, *FANCA* and *PALB2* genes. DDR wt: DDR wild type.

**Table 2 cancers-14-02950-t002:** Phase III trials with PARP inhibitors.

Type of Tumor	Author	Principal Endpoint	Treatment	Benefit	OS Benefit	Target	Sub-Population
Breast Cancer							
Localized disease	Tutt et al., 2021 [80]	DFS	Local treatment and neoadjuvant or adjuvant chemotherapy. Olaparib vs. placebo.	Yes	NS	*gBRCA1/2*	71.3% *BRCA1* 28.3% *BRCA2*
Pre-treated M1 or unresectable	Diéras 2020 [93]	PFS	Carbo, pacli ± veliparib	Yes	NS	*gBRCA1/2*	-
	Litton 2018 [78]	PFS	Chemo ^1^ vs. Talazoparib	Yes	NS	*gBRCA1/2*	-
	Robson 2017 [79]	PFS	Chemo ^1^ vs. olaparib	Yes	NS	*gBRCA1/2*	-
	O’Shaughnessy 2014 [94]	PFS and OS	Carbo, gem ± iniparib	Yes	Yes	Triple negative	
Ovarian Cancer							
1st line maintenance	Coleman 2019 [95]	PFS	Carbo, pacli ± veliparib	Yes	NR	Platinum sensitive	30% *BRCA*, 60% HRD
	Gonzalez-Martin 2019 [76]	PFS	Niraparib vs. placebo	Yes	NS ^2^	Platinum sensitive	30% *BRCA* 51% HRD
	Ray-Coquard 2019 [76]	PFS	Olaparib + Bevacizumab	Yes	NR	Platinum sensitive	30% *BRCA* 50% HRD
	Moore 2018 [74]	PFS	Olaparib vs. placebo	Yes	NS ^2^	*BRCA1/2* ^3^	
Platinum sensitive recurrence	Coleman 2017 [96]	PFS	Rucaparib vs. placebo	Yes	NS ^2^	Platinum sensitive	35% *BRCA* 60% HRD
	Pujade-Lauraine 2017 [97]	PFS	Olaparib vs. placebo	Yes	NS	*gBRCA1/2*	
	Mirza 2016 [72]	PFS	Niraparib vs. placebo	Yes	NS	Platinum sensitive	*BRCA* and non-*BRCA* cohorts
Pancreatic Cancer							
1st line maintenance	Golan 2019 [61]	PFS	Olaparib vs. placebo	Yes	NS ^2^	*gBRCA1/2* + platinum sensitive	
Prostate Cancer							
Pre-treated M1 CRPC	De Bono 2020 [83]	PFS in cohort A	Olaparib vs. AA/enza	Yes	NS	Somatic HRD by NGS 15 genes multi-panel	Cohort A: *BRCA + ATM* Cohort B: non-*BRCA/ATM*

^1^ Physician’s choice chemotherapy. ^2^ Immature data published. ^3^ Only two patients had somatic *BRCA1/2* mutation.

**Table 3 cancers-14-02950-t003:** Genetic testing recommendations for breast and/or ovarian cancer, exocrine pancreatic cancer and prostate cancer. National Comprehensive Cancer Network (NCCN) guidelines V2.2022, American Society of Clinical Oncology (ASCO) somatic genomic testing and European Society of Medical Oncology (ESMO) recommendations for the use of next-generation sequencing (NGS) in metastatic cancer.

	Breast and/or Ovarian Cancer	Exocrine Pancreatic Cancer	Prostate Cancer
Hereditary testing criteria	All patients diagnosed with epithelial ovarian cancer (including fallopian or peritoneal cancer). Any blood relative with a known pathogenic/likely pathogenic variant. Personal history of breast cancer with specific features: - ≤45 years. - 46–50 years with any: Unknown family history Multiple primary breast cancers (synchronous or metachronous) ≥1 close relative with breast, ovarian, pancreatic or prostate cancer at any age - ≥51 years: ≥1 close blood relative with any: breast cancer ≤50 years, or male breast cancer/ovarian/ pancreatic cancer any age, or metastatic, intraductal/cribiform histology, or high-or very high-risk group prostate cancer any age. ≥3 diagnoses of breast cancer in patient and/or close blood relatives ≥2 close blood relatives with breast or prostate cancer at any age. Any age: TNBC. ≥1 close relative with male breast cancer at any age Aid in systemic treatment decisions or adjuvant treatment decisions. - Ashkenazi Jewish ancestry	All individuals diagnosed. First-degree relatives of individuals diagnosed *	Metastatic prostate cancer Intraductal/cribriform histology High or very high-risk group Family history: - ≥1 close blood relative with breast cancer 50 years, or ovarian/pancreatic cancer any age, or metastatic, intraductal/cribriform histology, or high- or very-high risk prostate cancer. - ≥2 close blood relatives with breast or prostate cancer. - Ashkenazi Jewish ancestry
Genetic testing process			
-Familial pathogenic/likely pathogenic variant known	Testing for specific familial pathogenic/likely pathogenic variant	Testing for specific familial variant. If Ashkenazi Jewish descendent: test for all three-founder pathogenic/likely pathogenic variants.	Consider NGS panel testing.
-No known familial pathogenic/likely pathogenic variant	Comprehensive testing with multigene panel	Comprehensive testing with multigene panel.	In the abscense of family history or clinical features may be of low yield.
Germline recommendations	*BRCA1*, *BRCA2*, *ATM*, *BARD1*, *BRIP1*, *CDH1*, *CDKN2A*, *CHEK2*, *NBN*, *NF1*, *PALB2*, *PTEN*, *RAD51C*, *RAD51D*, *STK11*, *TP53.* Lynch syndrome genes *(MLH1*, *MSH2*, *MSH6*, *PM2).*	*BRCA1*, *BRCA2*, *ATM*, *CDKN2A*, *Lynch syndrome genes (MLH1*, *MSH2*, *MSH6*, *EPCAM)*, *PALB2*, *STK11 and TP53.*	*BRCA1*, *BRCA2*, *ATM*, *PABL2*, *CHECK2.* Lynch syndrome genes *(MLH1*, *MSH2*, *MSH6*, *PM2).* *HOXB13* may be valuable for family counselling.
Somatic testing ASCO recommendations	*BRCA 1/2* *NTRK1*, *NTRK2*, *NTRK3* fusions *MSI-H*, *TMB-H*		
	Breast cancer: *ERBB2* amplification. Oncogenic mutations in *PIK3CA* in HR+ HER2−. Ovarian cancer: GIS-positive or HRD-positive.		*MSI-H*, *ATM*, *BARD1*, *BRIP1*, *CDK12*, *CHEK1*, *CHEK2*, *FANCL*, *PALB2*, *RAD51B*, *RAD51C*, *RAD51D*, *RAD54L.*
ESMO Scale for clinical actionability of molecular targets	Metastatic breast cancer: *BRCA1/2* (germline/somatic)	Advanced pancreatic cancer: *BRCA1/2* germline/somatic mutations, *MSI-H*	Advanced prostate cancer: *BRCA1/2* somatic mutations/deletions, *MSI-H*, *ATM* mutations/deletions
NGS recommendations	Tumour multigene NGS can be used in ovarian cancer to determine somatic *BRCA1/2*. In breast cancer, no current indication for tumour multigene NGS.	No current indication for tumour multigene NGS	Multigene tumour NGS to assess level I alterations.

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
