# Peer review of "The Homologous Recombination Deficiency Scar in Advanced Cancer: Agnostic Targeting of Damaged DNA Repair"

_cancers, 2022, doi:10.3390/cancers14122950_

Round 1
Reviewer 1 Report
The manuscript entitled “The Homologous Recombination Deficiency Scar in Advanced Cancer: Agnostic Targeting of Damaged DNA Repair” described application of DNA repair targeted inhibitor for HRD associated tumor aiming synthetic lethality. The authors especially focus on BRCA1/BRCA2 genetic mutation associated tumors and PARP inhibitors. The paper is well documented and described about current situation of drug trial of DNA damage targeted inhibitor for HRD associated tumors. There is a merit for publish in Cancers.
Minor point
1. P1, abstract first term “BRCA1” might be regular font (not bold).
2. Figure 2A and 2B is separated, so, Figure 2 and Figure 3 are better to read, please re-number.
3. P6, paragraph 6, please define OS, this is first mention in text.
Reviewer 2 Report
This review manuscript introduced the relationship between HRD and cancer therapy in multiple tumors, and discussed the values of HRD scars in tumor treatment. This information is important for helping readers understand the current status of HRD-targeted cancer therapy. However, the manuscript writing needs to be improved to make information clear and avoid nonprofessional terms and description of DNA repair. Other comments are as follows:
1. Figure 1 is hard to understand, particularly the upper part. In addition, the sentence in Figure 1 legend, “HR system has functions in DNA repair, cell cycle regulation, transcriptional activation and chromatin remodeling” is not appropriate. HR functions in DNA repair, but does not function in other processes. Some proteins may have multiple functions. Thus, they may not only be involved in DNA repair, but also involved in cell cycle regulation, transcriptional activation and chromatin remodeling.
2. Th authors introduced the GIS score at the end of the manuscript. But it is unclear how the HRD scores are determined and calculated. What is the baseline of the HRD scores in normal tissues? Whether are GIS and HRD Scores the same or not? This information should be provided before discussing Figure 2B.
3. Full name should be listed for Acronym that appears at the first time, such as PSA, OS, PFS.
Round 2
Reviewer 2 Report
The authors added some sentences and words, but it appears that some previous sentences or words were not appropriately deleted. Please double check the manuscript to make sure that the modified parts are same as those appears in the Author's response.
